# Effect of umbilical cord blood stem cell transplantation on restenosis after endovascular interventional therapy for diabetic hindlimb vascular disease

**Hai-Xia Ding[1], Na Xing[1], Hong-Fang Ma[1], Lin Hou[1], Chao-Xi Zhou[2], Ya-Ping Du[1], Fu-Jun Wang[1]***

1 Department of Endocrinology, The Fourth Hospital of Hebei Medical University, Shijiazhuang, China,
2 Department of Gastrointestinal Surgery, The Fourth Hospital of Hebei Medical University, Shijiazhuang, China

* dhx-369@163.com

**Data Availability Statement:** There are acceptable restrictions to access to our data because data contain potentially sensitive information. The Ethics Committee of The Fourth Hospital of Hebei Medical

## Abstract

This study aimed to investigate the mechanism of human umbilical cord blood stem cell (HUCBSC) transplantation on restenosis after percutaneous transluminal angioplasty (PTA) for diabetic hindlimb vascular disease in rabbits. After successfully preparing a rabbit model of diabetic hindlimb vascular disease, 16 rabbits were randomly assigned to two groups. Of these, 8 rabbits received PTA surgery alone (PTA group), and the other 8 rabbits received PTA and HUCBSC (PTA+HUCBSC group) treatments. Five more healthy rabbits were set as healthy control (HC group). Samples were collected after 4 weeks of treatment. The expressions of regulator of calcineurin 1 (RCAN1) and calcineurin A (CnA) in the diseased artery were detected by immunofluorescence staining. The distribution of HUCBSCs was observed by pathological examination in transplanted artery, distal artery, and liver. Cytology experiments were applied to assess the levels of JAK and STAT3, and the migration and proliferation of human aortic vascular smooth muscle cells (HA-VSMC). In the rabbit model of diabetic vascular lesions in the hindlimbs, we found the stenosis of the femoral artery became more and more serious with time, and the expression level of PCNA positive cells was also gradually increased. The expression levels of RCAN1 and CnA in the PTA+HUCBSC group were significantly lower than those in PTA group. HUCBSC inhibited the migration and proliferation of HA-VSMC via JAK/STAT3 pathway. After HUCBSC local transplantation, HUCBSC had no distal tissue distribution. HUCBSC transplantation may prevent restenosis after PTA of diabetic hindlimb vascular disease through JAK/STAT3 pathway.

## Introduction

In recent years, the incidence of diabetic lower extremity arterial disease (LEAD) has increased year by year, and the results of epidemiological investigation show that the overall prevalence

University has imposed these restrictions on our study. Data requests can be made to our institution by calling +86-311 - 86095588.

**Funding:** The authors received no specific funding for this work.

**Competing interests:** The authors have declared that no competing interests exist.

of LEAD is 21.2% [1]. The risk of cardiovascular events is significantly increased in LEAD patients. Diabetic LEAD is not only a high risk factor for diabetic foot, but also an important reason for cardiovascular and cerebrovascular events in diabetic patients. Many characteristics of the interventional therapy, i.e. quick onset, small trauma, high success rate, and easy to perform, etc., make it an indispensable treatment for LEAD. However, in-stent restenosis (ISR) or non-stent restenosis after interventional therapy has become the main factor affecting its long-term prognosis [2]. Furthermore, other drawbacks after stent placement include delayed healing of the vascular endothelium, poor adhesion of the stent, thrombosis in the stent, stent fracture and distortion. Although the continuous advancement of balloon and stent technology has gradually reduced the rate of restenosis, restenosis after intervention is still a major problem that plagues clinicians.

Drug-coated balloon (DCB), a new concept and technology with the characteristics of "intervention without implantation" came into being [3]. With the development of volume reduction interventional devices such as plaque ablation, atherectomy, and microwave ablation, the concept of "intervention without implantation" will become the dominant mode of interventional therapy in the future, and drug-loaded devices will become the mainstream to reduce restenosis after intervention technology. DCB is a combination of plain old balloon angioplasty (POBA) and drug elution technology to attach drugs that inhibit vascular endothelial proliferation to the surface of the balloon. During the balloon expansion, the drugs are delivered to the local vascular wall to inhibit the proliferation of smooth muscle cells and prevent vascular restenosis [3, 4]. Up to the present, 67 cases of LEAD were treated with DCB in our center, and the patency rate of 10-month follow-up is 100%. However, no matter how the stent material is upgraded and the application of DCB, the occurrence of postoperative restenosis is still faced. The application of DCB still has some shortcomings, such as short drug release time, elastic retraction, and poor anti-proliferation effect [5, 6]. Therefore, the prevention and treatment of restenosis after interventional surgery has always been a medical problem to be solved urgently in the medical field.

A previous study has shown that ISR is associated with a range of inflammatory responses triggered by impaired endothelial function [7]. It is well known that stem cell transplantation can promote angiogenesis and protect vascular endothelial function. Umbilical cord blood is rich in hematopoietic stem cells, mesenchymal stem cells and endothelial progenitor cells, and is easy to obtain. A number of studies have shown that it has multiple differentiation potentials and can be used to treat diabetic peripheral neurovascular diseases [8–11]. In our previous study, a rabbit model of diabetic hindlimb vascular lesions was established. After interventional treatment, local injection of cord blood stem cells was performed in the transplantation group, which showed the intimal area and the ratio of intima area to media area in the dilated occlusion site were significantly decreased, indicating that human umbilical cord blood stem cell (HUCBSC) transplantation could effectively intervene the inflammatory response after endothelial injury and prevent restenosis [12]. However, the mechanism of inhibiting restenosis of HUCBSC after transplantation and whether HUCBSC inhibited restenosis in other organs after local transplantation are still unknown.

Regulator of calcineurin 1 (RCAN1) is an endogenous regulator of calcineurin. It interacts with calcineurin A (CnA) through its exon structural domain to regulate various pathophysiology related to CnA signaling pathway. RCAN1 plays a certain regulatory role in the proliferation of vascular smooth muscle cells, angiogenesis, atherosclerosis (AS), and vascular intimal hyperplasia. Nerea et al. [13] found that RCAN1 was significantly higher in the coronary tissues with AS plaques than in the non-atherosclerotic coronary arteries. This study intends to observe whether interventional therapy combined with HUCBSC transplantation can inhibit restenosis by changing the expression levels of RCAN1 and CnA.

The JAK-STAT family has 4 Janus kinase (JAK1-3, TYK2) and 7 STATs (1, 2, 3, 4, 5a, 5b, and 6) [14]. JAK-STAT signaling molecules can exist in AS plaques and vascular cells stimulated by inflammation. Hashimoto et al. [15] indicated that inhibition of the JAK/STAT signaling pathway may have an antagonistic effect on lipopolysaccharide-induced AS. Induced or simulated suppressor of cytokine signaling (SOCS) inhibit the development of AS by inhibiting the JAK/STAT signaling pathway [16]. Deficiency of STATs in vascular endothelial cells or inflammatory cells can inhibit the formation of AS plaques in mice, and JAK2 inhibitors can reduce the formation of angiogenesis intima. JAK/STAT3 signaling pathway also plays an important regulatory role in cell proliferation and apoptosis, and whether it is involved in the prevention of restenosis after HUCBSC transplantation has not yet been reported.

In order to provide further theoretical support for the prevention and treatment of restenosis after interventional therapy for clinical diabetic LEAD, this project will further observe the effect of cord blood stem cells on restenosis after interventional therapy for diabetic LEAD in rabbits. The possible downstream mechanism of HUCBSC transplantation for restenosis after interventional therapy in diabetic LEAD is to be studied in depth.

## Materials and methods

### Materials and reagent

The subjects of the study were 25 New Zealand healthy male white rabbits (Animal Experiment Center of Hebei Medical University), weighing (2.3 ± 0.3) kg. The density of lymphocyte separation solution is 1.077g/mL (Haoyang Company, Tianjin, China). Experimental reagents: lymphocyte separation solution, density 1.077g/mL (Haoyang Company, Tianjin, China), DAPI (Fu Bai Ke Biotechnology Co., Ltd., Beijing, China), RCAN1 and CnA rabbit anti-human polyclonal antibody, immunohistochemical kit (Huaxia Ocean Technology Co., Ltd., Beijing, China), p-Stat3-pAb, Western blotting reagents (CST Corporation, USA), human aortic vascular smooth muscle cells (HA-VSMC) and paclitaxel (Fusheng Industrial Co., Ltd., Shanghai, China), HUCBSC (Qilu Cell Bank, Shandong, China). The experimental equipments are as follows: RT-6000 enzyme label analyzer (Rayto Life and Analytical Sciences Co., Ltd., Shenzhen, China), 3mm x 40mm balloon (Medtronic, USA) and Inverted fluorescence microscope (Nikon Corporation, Japan).

### Methods

**Rabbit model of diabetic hind limb vascular disease.** Twenty rabbits were adaptively fed with basic for 1 week, and then fed with 100g high-fat daily. The high-fat formula: 2% cholesterol, 1% sodium deoxycholate, 5% lard, 92% ordinary feed. In the 5th week, all rabbits were fasted for 16 hours, and 60 mg/kg streptozocin (STZ) was injected rapidly through the periauricular vein. Fasting blood samples were collected once in the first week and the second week after STZ modeling. After blood collection, the serum was separated in time, and fasting blood glucose (FBG) was measured. FBG ≥ 11.0 mmol/L was judged as successful modeling of diabetes [17]. The study protocol was approved by the Ethics Committee of the Fourth Hospital of Hebei Medical University.

The rabbits were anesthetized with 3% pentobarbital 1mL/kg intravenously, and exposed the femoral artery at the pulse of the right inguinal artery. Gently ligate the femoral artery with 7 surgical suture avoiding the vascular branches. Then suture the incision and inject penicillin sodium 200,000 U/ rabbit intramuscularly. One week after surgery, bovine serum albumin (250mg/kg) was injected intravenously and the high-fat diet was continued. One rabbit was killed at 1 week, 2 weeks, 3 weeks, 4 weeks after modeling, and pathological sections were taken from the diseased femoral artery. After the successful modeling, the remaining 16 rabbits were randomly divided into two groups, 8 of which received percutaneous transluminal angioplasty (PTA) procedure

alone (PTA group), 8 underwent PTA combined with HUCBSC transplantation (PTA+-HUCBSC group), and 5 healthy rabbits were taken as healthy control group (HC group) at the same time. After 15–20 minutes of compression and hemostasis after intervention, the bandage was pressurized and bandaged for fixation. After 4–6 hours, the bandage was removed. The blood glucose and body temperature of the experimental rabbits were monitored twice a day, morning and evening. Whether the puncture point was bleeding, infection, and the color change of the lower limbs during the operation were observed. The eating, drinking and activities of the experimental rabbits were also observed. In this experiment, one rabbit developed a high body temperature greater than 49˚C, which was considered to be related to the skin infection at the surgical puncture point, and recovered after 3 days of anti-inflammatory administration of penicillin. Two rabbits suffered from lack of energy, which was considered to be hyperglycemia, and recovered after subcutaneous injection of small dose of insulin.The rabbits were euthanized by ear intravenous injection of pentobarbital sodium (100mg/kg) at the end of the experiment.

**Cell phenotype identification and labeling of HUCBSC.** After successfully resuscitating the HUCBSC, inoculate them in a 60×15mm petri dish (concentration $1×10^6$/mL), and place it in an incubator containing 5% $CO_2$ at 37˚C. The HUCBSC was cultured with low-sugar dulbecco's modified eagle medium (DMEM), and the medium was changed every day. After 7 days of culturation, if the cells adhere well and the cover 80–90% of the bottom area of the dish, discard the culture medium, rinse with phosphate buffer saline (PBS) three times, and take pictures of the adherent stem cells to observe their morphological presentation. Then added 0.25% trypsin-EDTA to digest the adherent stem cells, and the digestion was stopped when the cell gap increased under the microscope. The adherent cells were repeatedly blown to the wall with a sterile straw until they fell off. The cell suspension was placed in a centrifuge tube and centrifuged at a rate of 1000 R/min for 3 min. Next, add 2mL PBS solution to dilute, pipet evenly and transfer to EP tube, add 10uL monoclonal antibody CD34-PE and 10 monoclonal antibody CD45-PE to each EP tube, and stain for 30min in the dark at room temperature. After washing with PBS solution, add 1mL of PBS solution. The cell suspension in each EP tube is filtered with a 200-mesh disposable nylon membrane. Flow cytometry was then used to detect the expression of cell surface markers CD34 and CD45 to determine cell viability. The collected cells were washed with PBS and centrifuged for two times, and low-glucose DMEM was added to prepare cell suspension. Dil application solution of 5μL was added at a density of $1×10^9$/L, incubating at 37˚C for 25 min, and centrifuging at 1500 r/min for 5 min. The solution was then washed with PBS and centrifuged for two times, and added to low-sugar DMEM medium. The color of the cells was observed by fluorescence microscopy, and then the cells were cultured in an incubator at 37˚C with 5%$CO_2$. The fluorescence intensity and morphological changes of the cells at 24, 48, and 72 hours were observed.

**Interventional therapy and HUCBSC transplantation in rabbits with diabetic hindlimb vascular disease.** Eight diabetic hindlimb vascular disease rabbits were anesthetized intravenously with 3% pentobarbital (1mL/kg). Heparin (200U/kg) was injected into the ear vein at the beginning of the operation for anticoagulation. After successful puncture at the proximal femoral artery, 3×40mm PTA balloon was inserted into the femoral artery, inflated the balloon at 12 atmospheres for 3 minutes. Subsequently, the proximal and distal ends of the femoral artery were occluded with vascular clamps and then the balloon was withdrawn, the femoral artery was sutured, and the blood flow was restored. Rabbits in the PTA+HUCBSC group were injected with about 0.5mL of the prepared HUCBSC suspension (concentration $1×10^6$/mL) into the proximal femoral artery. Rabbits in the PTA group were injected with the same amount of PBS as a control. The puncture sites of all rabbits were sutured postoperatively. Heparin (200U/kg.d) was injected through the ear vein for anticoagulation, and penicillin was injected to prevent infection for 3 consecutive days.

**Histopathological examination.** At 1 week, 2 weeks, 3 weeks, and 4 weeks after modeling, the femoral artery of the right hindlimb was separated under intravenous anesthesia and vascular specimens were collected. After fixed with 4% paraformaldehyde, and embedded in paraffin, sections were performed, and the morphology of blood vessels was observed by HE staining. The morphology of elastic fibers was observed by EVG staining. Image-Plus6.0 image analysis software was used to measure the area of blood vessel lumen at the position of blood vessel plaque. Proliferating cell nuclear antigen (PCNA) and DAPI fluorescent staining were used to show the formation of atherosclerotic plaque and the degree of vascular wall stenosis.

After 4 weeks of treatment, the transplanted local arteries, distal arteries, liver and other tissues of the rabbits in the PTA+HUCBSC group were collected for histopathological examination to observe the distribution of cord blood stem cells.

**Immunofluorescence examination.** The expression of RCAN1 and CnA was detected by immunofluorescence examination of local transplanted arteries: samples were fixed with 4% paraformaldehyde. After sucrose gradient dehydration, frozen sections were performed. Then applied 0.01 mol/L citrate microwave antigen retrieval, 5% goat serum blocking, and added primary antibodies RCAN1 (1:50), SMA (1:100), CnA (1:100). Alexa Fluor 647-labeled goat anti-rabbit IgG (H+L) (1:50) and Cy3 labeled goat anti-rabbit IgG (H+L) (1:50) were added overnight at 4˚C. Finally, the samples were blocked by serum, stained with DAPI, and observed under a laser confocal microscope.

**MTS cell proliferation test.** In the logarithmic growth phase, HA-VSMC were seeded on 96-well culture plates with RPMI 1640 medium containing 10% FBS at a cell number of $2 \times 10^3$. They were divided into control group, cord blood stem cell group, and paclitaxel group. Added 10μL sterile MTS (500μg/mL) into the test hole, set 3 multiple holes at each time point, and test consecutively for 5 days. After adding MTS, incubate the cells at 37˚C for 2.5 hours, and place them in a microplate reader at 492nm wavelength to measure its optical density (OD) value. The experiment was repeated three times.

**Cell migration test.** Add 100μL ($1 \times 10^5$ cells/chamber) of the prepared HA-VSMC suspension to each upper chamber, and divide them into the control group, cord blood stem cell group, and paclitaxel group. Add 500μL of 10% fetal bovine serum medium to the chamber. After culturing for 48 hours, the Transwell chamber were took out. Then, the upper layer of the microporous membrane was wiped with a cotton swab, fixed with 4% paraformaldehyde, and stained with crystal violet for 30 minutes. The cells that migrated to the lower layer of the microporous membrane were counted under an inverted microscope. Next, each sample was counted from 10 fields of view, and the average number was calculated. The experiment was repeated three times.

**Western blot.** The co-cultured HA-VSMCs were lysed with RIPA lysate to extract protein, and the protein concentration was determined by bicinchoninic acid (BCA) kit. After the protein was electrophoresed by sodium dodecyl sulfate-polyacrylamide gel electrophoresis (SDS-PAGE), it was transferred to polyvinylidene fluoride (PVDF) membrane. The PVDF membrane was blocked with tris buffered saline tween (TBST) containing 3% bovine serum albumin (BSA), and the primary antibody was incubated overnight at 4˚C. The PVDF membrane was then took out, rinsed with TBST, incubated the secondary antibody at room temperature for 1 h, rinsed with TBST again. Then, the target protein signal was detected by substrate chemiluminescence ECL. The original uncropped and unadjusted images of western blot results were shown in S1 Fig.

## Statistical analysis

Differences between groups were analyzed using the analysis of variance (F test) and nonparametric test. Differences within groups were compared by t test. The level of statistical

significance for all the above tests was defined at a probability value of less than 0.05 (*P* < 0.05). All statistical analyses were performed using IBM SPSS Statistics *v*22.0 software (SPSS Inc., Chicago, IL, USA).

## Results

### Modeling of diabetic hind limb vascular disease in rabbits

After modeling, the rabbit vascular intima thickened and the vascular lumen gradually became stenosis. The formation of atherosclerotic plaques could also be observed. Immunofluorescence staining results showed that, the stenosis of femoral artery gradually increased and atherosclerotic plaques formed with the time (Fig 1).

### The inhibition of HUCBSC on HA-VSMC investigated by cytological experiments

Flow cytometry was used to detect the expression of cell surface markers CD34 and CD45 to determine cell viability. MTS cell proliferation test showed that both the HUCBSC group and the paclitaxel group could inhibit the proliferation of HA-VSMC. The proliferation rates of the two groups were 61.4±1.7% and 64.2±3.4%, respectively. The two group had a more significant inhibitory effect than normal control group (100.0±7.1%, *P*<0.01, respectively). The cell migration experiment showed that the number of cells that migrated in the HUCBSC group and paclitaxel group were 45.33±4.52/high power field (hfp) and 66.38±5.34/hpf, respectively, which were lower than the control group 108.45±8.45/hpf (*P*<0.01, respectively). Both groups

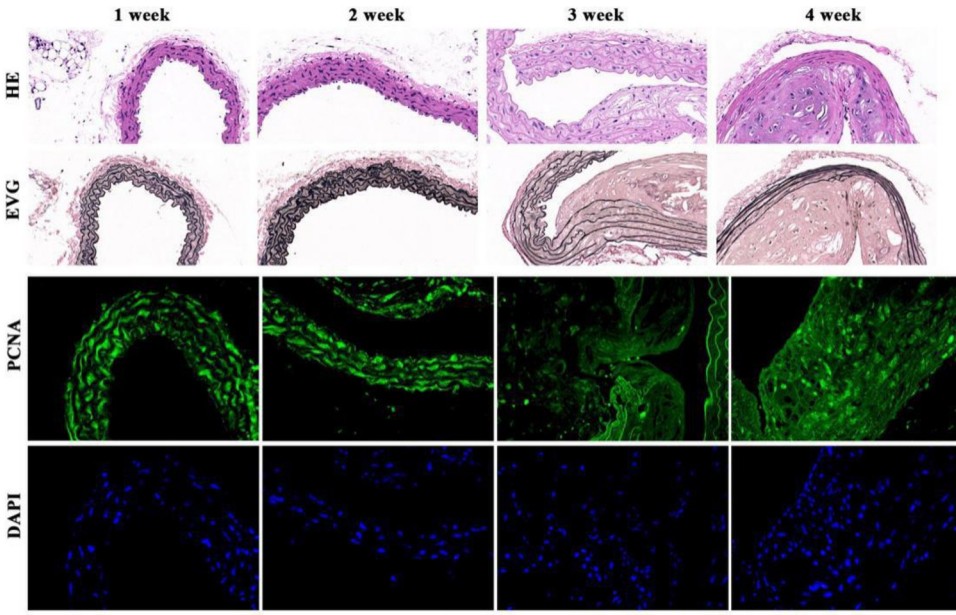

**Fig 1. Changes of vascular intima thickness and elastic fiber morphology of rabbits with diabetic hindlimb vascular lesions 1–4 weeks after modeling (x400).** First line: HE staining under ordinary light microscope showed that the atherosclerotic plaques gradually formed in the arterial wall and the lumen narrowed with time. Second line: EVG staining under ordinary light microscope showed that atherosclerotic plaques gradually formed in the arterial wall and the lumen narrowed with time. Third line: Fluorescence electron microscopy showed that PCNA positive cells in the arterial wall of rabbits increased and the arterial wall thickened gradually with time. Forth line: DAPI staining under fluorescence electron microscope showed that the proliferative cells in the arterial wall increased and the arterial wall thickened with time. HE: hematoxylin-eosin; EVG: Elastin Van Gieson; PCNA: proliferating cell nuclear antigen; DAPI: 4',6-diamidino-2-phenylindole.

could inhibit the migration of HA-VSMC. The inhibitory effect of HUCBSC group was stronger than that of paclitaxel group. Western blot showed that the p-STAT3 and p-JAK protein expression in HA-VSMCs in the HUCBSC group and paclitaxel group were significantly lower than that in the normal control group ($P<0.01$, respectively) (Fig 2).

## Effect of HUCBSC on restenosis of diabetic hind limb arteries in rabbits

Compared with the PTA group, the expression of RCAN1 and CnA in the arterial tissues of the PTA+HUCBSC group was significantly reduced, which was comparable to the normal group (Fig 3).

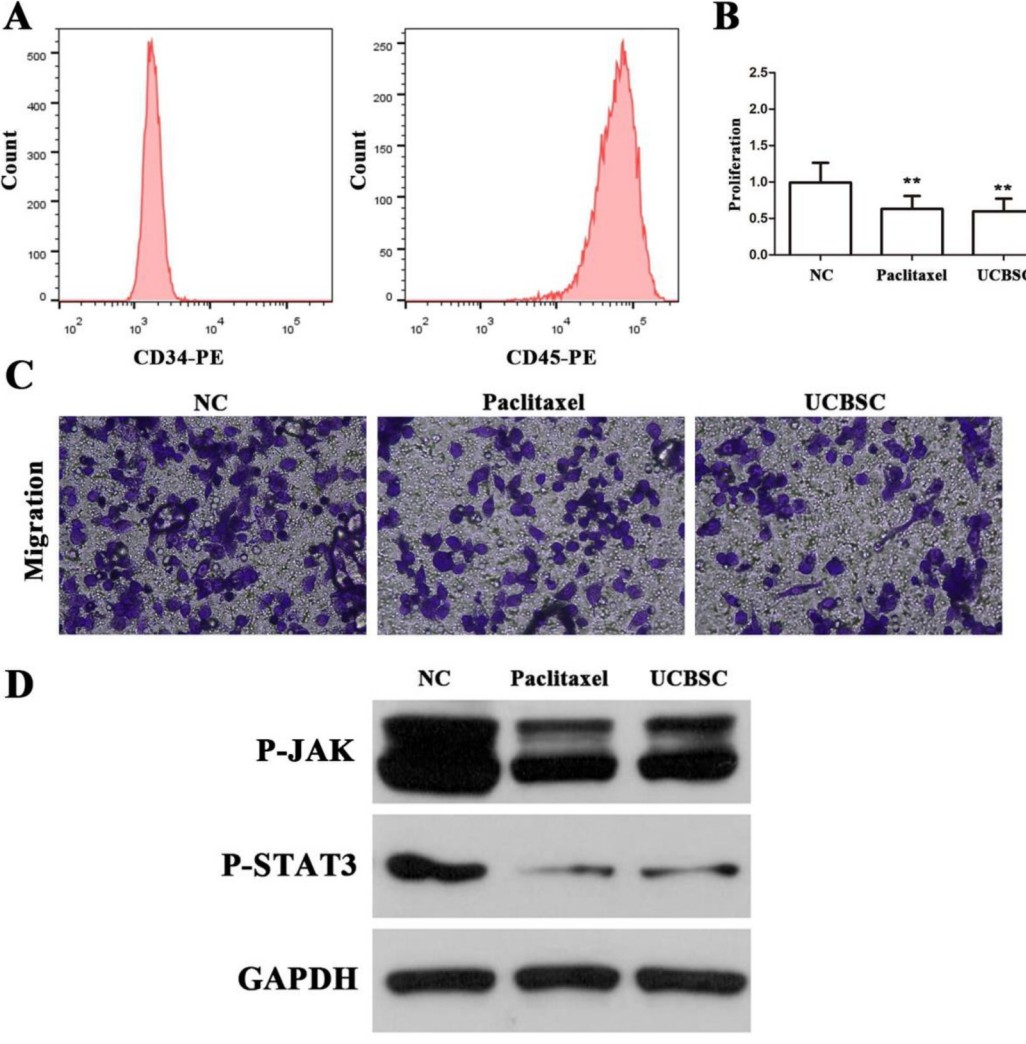

**Fig 2. Inhibitory effect of HUCBSC on HA-VSMC.** (A) Expression of HUCBSC cell surface markers CD34 and CD45 detected by flow cytometry. (B) The proliferation rate of each group compared by MTS cell proliferation test: The HUCBSC group and the paclitaxel group had a more significant inhibitory effect than the normal control group ($P<0.01$, respectively). **: $P<0.01$. (C) x400; The inhibition effect on HA-VSMC (cell migration experiment): The number of migrating cells in both HUCBSC group and paclitaxel group was lower than that in the control group, both groups could inhibit the migration of HA-VSMC cells, and HUCBSC group had a stronger inhibitory effect than paclitaxel group. (D) The expression of p-JAK and p-STAT3 protein in HA-VSMCs of each group: p-JAK and p-STAT3 protein expressions in HA-VSMCs were significantly decreased, both lower than those in control group ($P<0.01$).

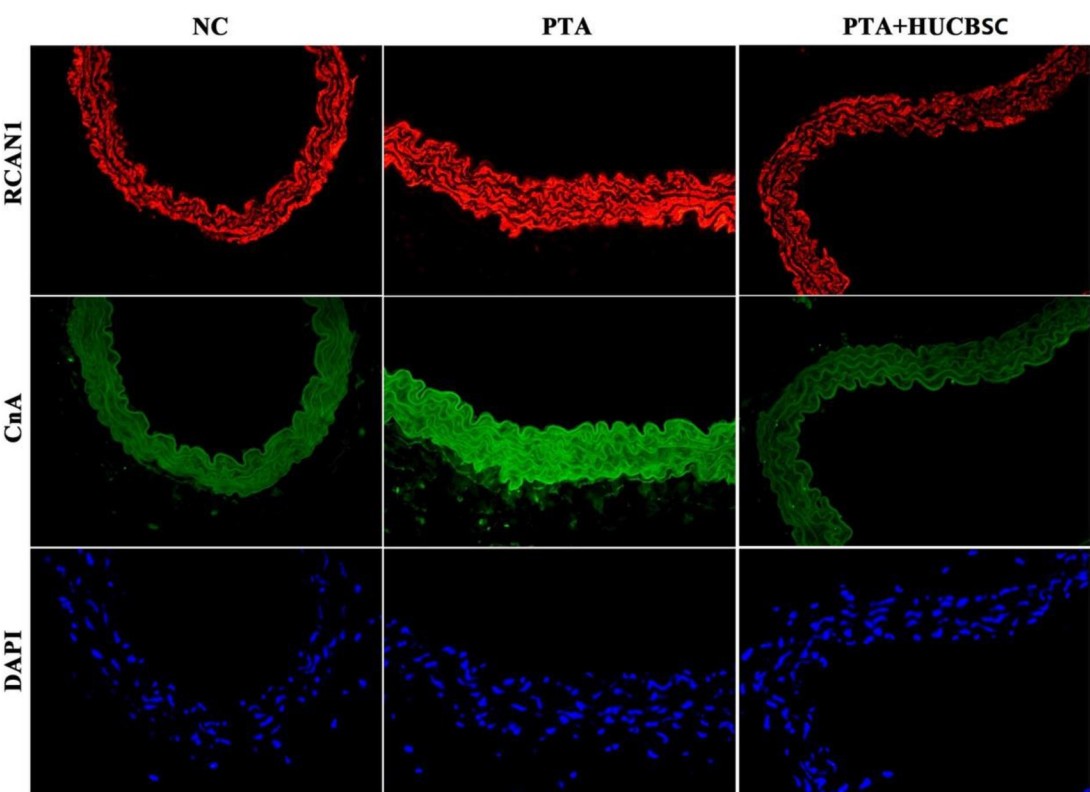

**Fig 3. Comparison of RCAN1 and CNA expression in each group under fluorescence microscope (x400).** First line: The expression of RCAN1in arterial wall under fluorescence electron microscope: the expression of RCAN1in the arterial tissues of the PTA+HUCBSC group was comparable to the normal group, and was lower than that of PTA group. Second line: The expression of CnA in arterial wall was observed under fluorescence electron microscope:the expression of CnA in the arterial tissues of the PTA+HUCBSC group was comparable to the normal group, and was lower than that of PTA group. Third line: DAPI staining of proliferating cells in the arterial wall under fluorescence electron microscope.

## Distribution of HUCBSC in local arteries, distal arteries and liver after transplantation

Pathological observation of the transplanted local arteries, distal arteries, and liver of the rabbits treated for 4 weeks showed that the HUCBSCs were distributed in the transplanted local arteries, but not in the distal arteries and liver (Fig 4).

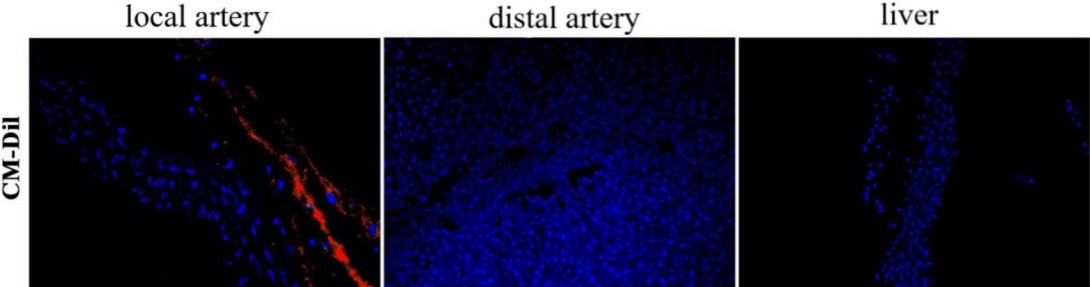

**Fig 4. The distribution of HUCBSC in local artery, distal artery and liver.** A small amount of HUCBSC can be seen in local artery (red), but not in distal artery or liver.

## Discussions

Interventional therapy of diabetic LEAD has become the main method of treatment due to its advantages such as small trauma, high success rate, good curative effect, and easy to perform. However, stents or intravascular restenosis after intervention seriously affect its long-term efficacy. The basic mechanism of ISR includes the aggregation of platelets, white blood cells, and macrophages, which further leads to the migration and proliferation of smooth muscle cells. The inflammatory response runs through the entire process of AS. Park SJ et al. [7] found that new in-stent AS may be the main reason for the formation of ISR. Therefore, how to prevent and reduce the inflammation and the occurrence of new AS after interventional surgery is the key to prevent restenosis. Stem cell transplantation for the treatment of diabetic LEAD has been widely carried out worldwide and has achieved good results [18, 19]. Stem cells are mainly derived from bone marrow, peripheral blood and cord blood. HUCBSC has abundant resources, weak antigenicity, and low immune rejection, making it the first choice for allogeneic stem cell transplantation. We have made a rabbit diabetic hindlimb vascular disease model and injected HUCBSC locally for transplantation after interventional treatment. It was found that the risk factors for restenosis, VE-cadherin and Lp-PLA2, were significantly lower than those in the interventional treatment group. However, there was no significant difference compared with the normal control group. Local intima of femoral artery showed mild hyperplasia [19].

RCAN1 is an endogenous regulator of calcineurin. RCAN1 binds to CnA through its exon domain and regulates various pathophysiology related to CnA signaling pathway. CnA is a catalytic subunit of calcineurin (CaN) that can bind to calmodulin. Another regulatory subunit of CaN, calcineurin subunit B (CnB), can bind to calcium [20]. Regarding the different opinions about the effect of RCAN1 on CaN, Ma et al. [21] in 2015 further demonstrated the inhibitory effect of RCAN1 on CaN from the aspects of enzymology and thermodynamics. That is, the expressed RCAN1 protein preferentially binds to activated CaN. Its shared C-terminal 140–197 amino acids interact with the A subunit (CnA) of CaN to inhibit its activity and the transcription and expression of downstream genes, forming negative feedback regulation. However, some studies have found that the effect of RCAN1 on CaN can also be a promoting effect, so RCAN1 is called a regulator of CaN, not just an inhibitor [22]. Hilioti Z et al. [23] also confirmed that RCAN1 protein may inhibit CaN at a certain concentration and activate the CaN signaling pathway at another concentration. Another study indicated that the regulation of RCAN1 on CaN was related to the subtype of RCAN1 itself [24]. The results of this study showed that after HUCBSC transplantation, the expression levels of RCAN1 and CnA showed consistent changes, indicating that RCAN1 may promote the CaN signaling pathway.

It was found that RCAN1 plays a certain regulatory role in vascular smooth muscle cell proliferation, angiogenesis, atherosclerotic formation, intimal hyperplasia and other processes. AS is the main cause of myocardial ischemia, stroke and peripheral vascular occlusive disease. It is currently believed that RCAN1 protein may play a catalytic role in the early formation of AS. Studies have shown that RCAN1 is a potential AS-causing factor. Méndez-Barbero et al. [13] found that RCAN1 was significantly higher in coronary artery tissue with AS plaque than in coronary artery without AS. The results of this study showed that the expression levels of RCAN1 and CnA in the lesions of the PTA+HUCBSC group were significantly lower than those of the PTA group, which was consistent with the results of the above studies.

The JAK-STAT family has 4 Janus kinase (JAK1-3, TYK2) and 7 STATs (1, 2, 3, 4, 5a, 5b, and 6) [14]. JAK-STAT signaling molecules can exist in AS plaques and vascular cells stimulated by inflammation. Hashimoto et al. [15] indicated that inhibition of the JAK/STAT signaling pathway may have an antagonistic effect on lipopolysaccharide-induced AS. Induced or

simulated suppressor of cytokine signaling (SOCS) inhibit the development of AS by inhibiting the JAK/STAT signaling pathway [15].

This study found that HUCBSC may inhibit the migration and proliferation of HA-VSMC through the JAK/STAT3 pathway. Recent studies have shown that there may be six signaling pathways related to the intervention of AS inflammatory response, namely, Janus kinase/signal transducer and activator of transcription (JAK/STAT) signaling pathway, nuclear factor-κB (NF-κB) signaling pathway, mitogen-activated protein kinase (MAPK) dependent signaling pathway, reactive oxygen species (ROS) dependent signaling pathway, CD40-CD40L signaling pathway, Toll-like receptor (TLR) dependent signaling pathway and Hippo Pathway [25–27]. There may be other signaling pathway involved in the mechanism of preventing restenosis after HUCBSC transplantation, and further research is needed.

HUCBSC has the ability to automatically homing to injury and inflammatory regions. Homing is a multi-step coordinated process involving many cytokines, chemokines, adhesion factors and extracellular matrix degradation proteases. Different chemokines can be expressed locally after tissue injury: such as CXCL 4, CXCL12, and CXCL6. These chemokines are necessary for HUCBSC homing, and the local microenvironment is the initiating factor of homing. HUCBSC secretes various angiogenesis-promoting cytokines such as VEGF, bFGF and PEGF through paracrine mode. When ischemia and hypoxia occur in tissues, chemokines of inflammatory factors, such as IL-1, IL-6 and IL-7, can promote the migration of human umbilical cord-mesenchymal stem cells and homing to the damaged sites. Vascular cell adhesion factor 1, delayed antigen and soluble P-selectin expressed by HUCBSC help to adhere to injured vessel. The regeneration and extension of blood vessels require matrix metalloproteinases (MMPs) to decompose extracellular matrix (ECM), which is also conducive to the migration of endothelial cells. HUCBSC can secrete a variety of proteases, regulate and degrade the extracellular matrix, thereby facilitating its homing. In vitro culture found that MMP-2 and MT1-MMP are significantly related to the ability of HUCBSC to migrate to the endothelium [28]. The homing mechanism of HUCBSC is affected by many factors, such as the location of the injured tissue, the number of transplants, the way and timing of transplantation. The results of this study showed that Dil-labeled HUCBSC was distributed in the local arteries of transplantation, but not in the distal arteries and liver, and no tumorigenicity was found.

In conclusion, intervention treatment combined with HUCBSC transplantation may prevent in-stent restenosis through JAK/STAT3 pathway. After HUCBSC local transplantation, no distribution of distal tissues and no tumorigenicity was found. It provides a new method for the prevention and treatment of restenosis after intervention in diabetic LEAD patients.

## Supporting information

**S1 Fig. The original uncropped and unadjusted images of western blot results.**
(TIF)

## Author Contributions

**Conceptualization:** Hai-Xia Ding, Na Xing, Ya-Ping Du, Fu-Jun Wang.

**Data curation:** Hai-Xia Ding, Na Xing, Hong-Fang Ma, Lin Hou, Chao-Xi Zhou, Fu-Jun Wang.

**Formal analysis:** Hai-Xia Ding, Na Xing, Hong-Fang Ma, Lin Hou, Chao-Xi Zhou, Ya-Ping Du.

**Supervision:** Fu-Jun Wang.

**Writing – original draft:** Hai-Xia Ding, Na Xing, Fu-Jun Wang.

**Writing – review & editing:** Ya-Ping Du, Fu-Jun Wang.

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
