## [Decision Letter · Decision Letter 0]

14 Apr 2021

PONE-D-21-05884

Effect of umbilical cord blood stem cell transplantation on restenosis after endovascular interventional therapy for diabetic hindlimb vascular disease

PLOS ONE

Dear Dr Wang

Thank you for submitting your manuscript to PLOS ONE. After careful consideration, we feel that it has merit but does not fully meet PLOS ONE’s publication criteria as it currently stands. Therefore, we invite you to submit a revised version of the manuscript that addresses the points raised during the review process.

We look forward to receiving your revised manuscript.

Kind regards,

Xianwu Cheng, M.D., Ph.D., FAHA

Academic Editor

PLOS ONE

Additional Editor Comments:

As the authors gather from the reviews, the referee identified substantive methodological and statistical problems as well as several data presentation and the conclusions are very confusing and not clear.

Journal Requirements:

2. At this time, we request that you  please report additional details in your Methods section regarding animal care, as per our editorial guidelines:

(1) Please state the source of the rabbits used in the study (where they were purchased)

(2) Please include the method of euthanasia

(3) Please describe the post-operative care received by the animals, including the frequency of monitoring and the criteria used to assess animal health and well-being.

Thank you for your attention to these requests.

3. To comply with PLOS ONE submissions requirements, please provide the method of euthanasia in the Methods section of your manuscript.

4. Please provide additional information about each of the cell lines used in this work, including any quality control testing procedures (authentication, characterisation, and mycoplasma testing). For more information, please see http://journals.plos.org/plosone/s/submission-guidelines#loc-cell-lines.

"no funding receive."

7. PLOS ONE now requires that authors provide the original uncropped and unadjusted images underlying all blot or gel results reported in a submission’s figures or Supporting Information files. This policy and the journal’s other requirements for blot/gel reporting and figure preparation are described in detail at https://journals.plos.org/plosone/s/figures#loc-blot-and-gel-reporting-requirements and https://journals.plos.org/plosone/s/figures#loc-preparing-figures-from-image-files. When you submit your revised manuscript, please ensure that your figures adhere fully to these guidelines and provide the original underlying images for all blot or gel data reported in your submission. See the following link for instructions on providing the original image data: https://journals.plos.org/plosone/s/figures#loc-original-images-for-blots-and-gels.

8. PLOS requires an ORCID iD for the corresponding author in Editorial Manager on papers submitted after December 6th, 2016. Please ensure that you have an ORCID iD and that it is validated in Editorial Manager. To do this, go to ‘Update my Information’ (in the upper left-hand corner of the main menu), and click on the Fetch/Validate link next to the ORCID field. This will take you to the ORCID site and allow you to create a new iD or authenticate a pre-existing iD in Editorial Manager. Please see the following video for instructions on linking an ORCID iD to your Editorial Manager account: https://www.youtube.com/watch?v=_xcclfuvtxQ

Reviewers' comments:

Reviewer's Responses to Questions

**Comments to the Author**

1. Is the manuscript technically sound, and do the data support the conclusions?

Reviewer #1: Partly

Reviewer #2: Yes

2. Has the statistical analysis been performed appropriately and rigorously? 

Reviewer #1: I Don't Know

Reviewer #2: Yes

3. Have the authors made all data underlying the findings in their manuscript fully available?

Reviewer #1: Yes

Reviewer #2: Yes

4. Is the manuscript presented in an intelligible fashion and written in standard English?

Reviewer #1: Yes

Reviewer #2: No

5. Review Comments to the Author

Reviewer #1: 1- Introduction- sentence structure is awkward (...due to its advantages such as small trauma and quick effect.)- consider revising

2-Introduction- I agree that percutaneous intervention has become indispensible for treating PAD; however, increasingly providers are using less indwelling stents and/or are using drug-eluting technologies to ameliorate the risk of neointimal hyperplasia- can the authors provide more context for this evolving practice pattern in their introduction?

3-Introduction- regarding the sentence: "After interventional treatment, local injection of cord blood stem cells...between intimal area and medium film area..." please clarify.

4-Introduction- can the authors provide clarification for why they targeted RCAN1 specifically for in-stent restenosis? There are a variety of other pathways that are implicated in neointimal hyplerplasia proliferation. I understand that there are multi-step regulatory check points that this targets but why this and not another molecule?

5-Methods- I appreciate that the authors have catalogued the materials used in the experiments; however, it is difficult to interpret without its appropriate context- is there a way to integrate the information offered in the 1st paragraph into subsequent aspects of the methods so this information doesn't sit unsupported or without specific context to an experiment?

6-Methods- STZ model, can the authors provide a reference for the 11.0mmol/L threshold to corroborate the diabetic state diagnosis?

7-Methods- rabbit model- if I am reading the manuscript correctly, rabbits underwent 'gentle ligation' of the femoral artery using 7-0 suture; however, the subsequent paragraph discusses 'successful remodeling' and then subsequent angioplasty -- what was being angioplastied if the femoral artery was surgically ligated? Also, in-stent restenosis was discussed in the introduction but angioplasty alone was used, was there a group of animals that received an intravascular stent?

8-Methods- cell phenotype identification- several sentences use non-scientific/peer reviewed publication appropriate nomenclature to define what occurred- would consider revision (e.g. 'sucked into the centrifuge', etc.).

9-Methods- Interventional therapy and HUCBSC- can the authors provide a study table and/or flow chart that outlines how the experiments were conducted?

10-Methods- there is a description of placing the balloon in the femoral artery but also proximal and distal occlusion with vascular clamps- how can the clamps occlude the vessel proximally and distally while the balloon is in place, wont the artery have the shaft of the balloon in it so one of the vessels cannot be fully occluded?

11- Discussion- the authors highlight stents in the discussion; however, they only present experiments for plain balloon angioplasty- please clarify the overlap since mechanisms for in-stent restenosis and neointimal hyperplasia at balloon angioplasty sites are somewhat different.

Reviewer #2: In this manuscript, the authors provided evidence showing that the mechanism of human umbilical cord blood stem cell (HUCBSC) transplantation on the restenosis of percutaneous transluminal angioplasty (PTA) in the diabetic hindlimb vascular disease via RCAN1, Can and JAK/STAT3 pathway. In PTA+HUCBSC group, the expression levels of RCAN1 and CnA were significantly lower than those in PTA group and the HUCBSC inhibited the migration and proliferation of HA-VSMC and suppressed the levels of JAK and STAT3.

The following shortcomings need to be addressed to strengthen the study.

1. Why use the paclitaxel (an anti-cancer agent) as a control group in the cell experiments and how many doses were used in the HA-VSMC cell lines.

2. Dil-labeling kit was used to detect the HUCBSC, detailed information of procedure should be provided.

3. The authors did not provide the methods of HUCBSC treatment in vitro, does it have co-culture of HUCBSC and HA-VSMC cells?

4. In Fig 1 and Fig3 legends fluorescence electron microscope was employed, but there are no electron microscope images in the paper, please check it.

5. In Fig 4, please provide more detailed information of methods for better understanding.

6. No. 3, 4, 5, 6, 13, and 22 of references are in Chinese, please address.

7. No. 11 of reference is showed “INVALID CITATION”, please check it.

6. PLOS authors have the option to publish the peer review history of their article (what does this mean?). If published, this will include your full peer review and any attached files.

Reviewer #1: No

Reviewer #2: No

---

## [Author Response · Author response to Decision Letter 0]

23 Jun 2021

Review Comments to the Author

Reviewer #1: 1- Introduction- sentence structure is awkward (...due to its advantages such as small trauma and quick effect.)- consider revising

Reply: Thanks for your comment. The sentence has been revised as follows:

Many characteristics of the interventional therapy, i.e. quick onset, small trauma, high success rate, and easy to perform, etc., make it an indispensable treatment for LEAD. (Page 3, Line 56-58)

2-Introduction- I agree that percutaneous intervention has become indispensible for treating PAD; however, increasingly providers are using less indwelling stents and/or are using drug-eluting technologies to ameliorate the risk of neointimal hyperplasia- can the authors provide more context for this evolving practice pattern in their introduction?

Reply: Thanks for your comments. Relative context have been added in the Introduction section. (Page 3-4, Line 60-62, 65-82)

3-Introduction- regarding the sentence: "After interventional treatment, local injection of cord blood stem cells...between intimal area and medium film area..." please clarify.

Reply: Thanks for your comments. The sentence has been revised as follows:

After interventional treatment, local injection of cord blood stem cells was performed in the transplantation group, which showed the intimal area and the ratio of intima area to media area in the dilated occlusion site were significantly decreased.(Page 4, Line 92-93)

4-Introduction- can the authors provide clarification for why they targeted RCAN1 specifically for in-stent restenosis? There are a variety of other pathways that are implicated in neointimalhyplerplasia proliferation. I understand that there are multi-step regulatory check points that this targets but why this and not another molecule?

Reply: Thanks for your professional comments. Atherosclerosis is the main cause of myocardial ischemia, cerebral apoplexy and peripheral vascular occlusion.It has been suggested that RCAN1 may play a catalytic role in the early formation of atherosclerosis. Nerea et al. found that RCAN1 was significantly higher in the coronary tissues with AS plaques than in the non-atherosclerotic coronary arteries. (Page 4, Line 103-105) 

5-Methods- I appreciate that the authors have catalogued the materials used in the experiments; however, it is difficult to interpret without its appropriate context- is there a way to integrate the information offered in the 1st paragraph into subsequent aspects of the methods so this information doesn't sit unsupported or without specific context to an experiment?

Reply: Thanks for your comments. The reason why the“Materials and Reagent” section is placed in the first paragraph of the method part is that we hope to provide information on the manufacturers, specifications and models of the key instruments and reagents involved in the research to show the readers the source of these materials. As a paper involving animal research, we generally state the source information of these materials in the first paragraph of the method. The source information of these materials is only used as a description of information, not the focus of this research. The key point we want to describe is still the operational details of our experimental method. We are very grateful for your questions. 

6-Methods- STZ model, can the authors provide a reference for the 11.0mmol/L threshold to corroborate the diabetic state diagnosis?

Reply: Thanks for your comments. The reference has been added. We have observed in preliminary experiments that if the fasting blood glucose of rabbits was over 16 mmol/L, the postdietary blood glucose would be as high as 20 mmol/L. In this case, rabbits would show sluggishness and anorexia, increase the risk of ketoacidosis, and had a high mortality rate and high infection rate after intervention. They were not suitable for the long-term follow-up observation of this experiment. Therefore, FBG≥11.0 mmol/L was considered to be a successful model of diabetic rabbit.

7-Methods- rabbit model- if I am reading the manuscript correctly, rabbits underwent 'gentle ligation' of the femoral artery using 7-0 suture; however, the subsequent paragraph discusses 'successful remodeling' and then subsequent angioplasty -- what was being angioplastied if the femoral artery was surgically ligated? Also, in-stent restenosis was discussed in the introduction but angioplasty alone was used, was there a group of animals that received an intravascular stent?

Reply: Thanks for your comments. The process of gentle ligation you mentioned is the process of vascular restenosis in this study. The rabbits that were determined to be successful in diabetic modeling exposed the femoral artery at the right inguinal pulsation, and gently ligated the femoral artery with a 7-0 suture at the avascular branch, taking care not to ligate completely. Therefore, this diabetic hindlimb vascular stenosis model rabbit was successfully made.After the successful remodeling, the remaining 16 rabbits were randomly divided into two groups, 8 of which received percutaneous transluminal angioplasty（PTA） procedure alone (PTA group), 8 underwent PTA combined with HUCBSC transplantation (PTA+HUCBSC group). The rabbits in the PTA group and the PTA+HUCBSC group were treated with interventional therapy.

8-Methods- cell phenotype identification- several sentences use non-scientific/peer reviewed publication appropriate nomenclature to define what occurred- would consider revision (e.g. 'sucked into the centrifuge', etc.).

Reply: Thanks for your comments. The sentence has been revised as follows:

The cell suspension was placed in a centrifuge tube and centrifuged at a rate of 1000 R/min for 3 min. (Page 7, Line 174-175)

9-Methods- Interventional therapy and HUCBSC- can the authors provide a study table and/or flow chart that outlines how the experiments were conducted?

Reply: Thanks for your comments. The flow chart is supplied below.

10-Methods- there is a description of placing the balloon in the femoral artery but also proximal and distal occlusion with vascular clamps- how can the clamps occlude the vessel proximally and distally while the balloon is in place, wont the artery have the shaft of the balloon in it so one of the vessels cannot be fully occluded?

Reply: Thanks for your comments. Because the femoral artery of rabbits is limited in length, the shortest balloon length used in the experiment was 40mm. During expansion, part of the balloon was outside the blood vessel while the other part was inside, and the balloon located inside the blood vessel played a role of expansion. In this way, clamping both ends of the femoral artery would not cause the balloon to be occluded. However, due to the large wound surface of the blood vessel puncture in this operation, the puncture point of the femoral artery needs to be sutured. See the figure below.

11- Discussion- the authors highlight stents in the discussion; however, they only present experiments for plain balloon angioplasty- please clarify the overlap since mechanisms for in-stent restenosis and neointimal hyperplasia at balloon angioplasty sites are somewhat different.

Reply: Thanks for your comments.This experiment simulates an intervention on the mechanism of restenosis after intervention, which involves the problem of intimal hyperplasia. Of course, the mechanism of stenosis also includes smooth muscle hyperplasia, changes in the media, and the formation of atherosclerosis. This study focuses on the mechanism of restenosis after plain old balloon angioplasty and restenosis after stent placement, so in-stent restenosis was mentioned in the discussion.

Reviewer #2: In this manuscript, the authors provided evidence showing that the mechanism of human umbilical cord blood stem cell (HUCBSC) transplantation on the restenosis of percutaneous transluminal angioplasty (PTA) in the diabetic hindlimb vascular disease via RCAN1, Can and JAK/STAT3 pathway. In PTA+HUCBSC group, the expression levels of RCAN1 and CnA were significantly lower than those in PTA group and the HUCBSC inhibited the migration and proliferation of HA-VSMC and suppressed the levels of JAK and STAT3.

The following shortcomings need to be addressed to strengthen the study.

1. Why use the paclitaxel (an anti-cancer agent) as a control group in the cell experiments and how many doses were used in the HA-VSMC cell lines.

Reply: Thanks for your comments. Paclitaxel coated balloon angioplasty is gradually applied to the treatment of vascular proliferative diseases, such as restenosis after coronary stenting and diabetic peripheral vascular disease. And clinical studies have confirmed that the paclitaxel balloon stent is safe and effective in reducing coronary restenosis and improving limb ischemia. Therefore, paclitaxel was used as a control group. 

After paclitaxel was completely dissolved in absolute ethanol, the culture solution was diluted to 1mmol/L, then filtered and sterilized with a 0.22µm filter, and stored in the refrigerator at -4℃ for later use. The pre-prepared paclitaxel solution was added to the culture medium and diluted to the required concentration before use. This experiment used a low concentration of paclitaxel (working concentration was 100nM), which was non-cytotoxic.

2. Dil-labeling kit was used to detect the HUCBSC, detailed information of procedure should be provided.

Reply: Thanks for your comments. The detailed information of procedure are as follows:

The collected cells were washed with PBS and centrifuged for two times, and low-glucose DMEM was added to prepare cell suspension. Dil application solution of 5µL was added at a density of 1×109/L, incubating at 37℃ for 25 min, and centrifuging at 1500 r/min for 5 min. The solution was then washed with PBS and centrifuged for two times, and added to low-sugar DMEM medium. The color of the cells was observed by fluorescence microscopy, and then the cells were cultured in an incubator at 37 ℃ with 5%CO2. The fluorescence intensity and morphological changes of the cells at 24, 48, and 72 hours were observed. (Page 7, Line 181-188)

3. The authors did not provide the methods of HUCBSC treatment in vitro, does it have co-culture of HUCBSC and HA-VSMC cells?

Reply: Thanks for your comments. During the MTS cell proliferation test and cell migration test, HUCBSC and HA-VSM were co-cultured in HUCBSC group. Detailed information were shown in these two section. (Page 9, Line 227-244)

4. In Fig 1 and Fig3 legends fluorescence electron microscope was employed, but there are no electron microscope images in the paper, please check it.

Reply: Thanks for your comments. The Fig 1 and Fig3 have been addd as follows:

Fig 1: Changes of vascular intima thickness and elastic fiber morphology of rabbits with diabetic hindlimb vascular lesions 1-4 weeks after modeling (x400).

Fig 3：Comparison of RCAN1 and CNA expression in each group under fluorescence microscope (x400)

5. In Fig 4, please provide more detailed information of methods for better understanding.

Reply: Thanks for your comments. My understanding of this question is to provide more details of immunofluorescence examination. The detailed information of methods was supplied in the Methods section. (Page 8, Line 218-226)

6. No. 3, 4, 5, 6, 13, and 22 of references are in Chinese, please address.

Reply: We are deeply sorry for this error, and we have modified the references.

7. No. 11 of reference is showed “INVALID CITATION”, please check it.

Reply: We are deeply sorry for this error, and we have modified the references.

Round 2, Editor comments:

1) We noted that you have not responded to our earlier request regarding your ORCID iD. Please note that PLOS ONE requires that all corresponding authors have an authorised ORCID iD. As we previously mentioned, we cannot proceed with the publication process until this request has been met. At this time, please link your ORCID ID to your PLOS Editorial Manager profile.

Reply: I have fitched the ORCID number in my personal information page.

2) Thank you for providing the following information regarding data availability in your response to reviewers:

"Thanks for your comments. The datasets generated and analyzed during the current study are available from the corresponding author on reasonable request."

We note that you have indicated that data from your study are available upon request to the author.

Reply: There are acceptable restrictions to access to our data because data contain potentially sensitive information, however, the datasets generated and analyzed during the current study are available from the corresponding author on reasonable request.

---

## [Decision Letter · Decision Letter 1]

12 Jul 2021

Effect of umbilical cord blood stem cell transplantation on restenosis after endovascular interventional therapy for diabetic hindlimb vascular disease

PONE-D-21-05884R1

Dear Dr Wang

We’re pleased to inform you that your manuscript has been judged scientifically suitable for publication and will be formally accepted for publication once it meets all outstanding technical requirements.

Kind regards,

Xianwu Cheng, M.D., Ph.D., FAHA

Academic Editor

PLOS ONE

Additional Editor Comments (optional):

All original concerns have been addressed by the authors.

Reviewers' comments:

Reviewer's Responses to Questions

**Comments to the Author**

1. If the authors have adequately addressed your comments raised in a previous round of review and you feel that this manuscript is now acceptable for publication, you may indicate that here to bypass the “Comments to the Author” section, enter your conflict of interest statement in the “Confidential to Editor” section, and submit your "Accept" recommendation.

Reviewer #1: All comments have been addressed

Reviewer #2: All comments have been addressed

2. Is the manuscript technically sound, and do the data support the conclusions?

Reviewer #1: Yes

Reviewer #2: Yes

3. Has the statistical analysis been performed appropriately and rigorously? 

Reviewer #1: Yes

Reviewer #2: Yes

4. Have the authors made all data underlying the findings in their manuscript fully available?

Reviewer #1: Yes

Reviewer #2: Yes

5. Is the manuscript presented in an intelligible fashion and written in standard English?

Reviewer #1: Yes

Reviewer #2: Yes

6. Review Comments to the Author

Reviewer #1: The authors have provided satisfactory responses to all my questions and comments. I have no further questions.

Reviewer #2: Methods: the samples ..…..observed under a laser confocal microscope.

Fig 1 Third line: Fluorescence electron microscopy showed…...

Fig 3 First line: …...under fluorescence electron microscope.

but there are no electron microscope images in the Figs, please check it.

7. PLOS authors have the option to publish the peer review history of their article (what does this mean?). If published, this will include your full peer review and any attached files.

Reviewer #1: No

Reviewer #2: No

---

## [Editor Report · Acceptance letter]

2 Aug 2021

PONE-D-21-05884R1 

Effect of umbilical cord blood stem cell transplantation on restenosis after endovascular interventional therapy for diabetic hindlimb vascular disease 

Dear Dr. Wang:

I'm pleased to inform you that your manuscript has been deemed suitable for publication in PLOS ONE. Congratulations! Your manuscript is now with our production department. 

Kind regards, 

on behalf of

Associate Prof. Xianwu Cheng 

Academic Editor

PLOS ONE